# Disentangled-Multimodal Privileged Knowledge Distillation for Depression Recognition with Incomplete Multimodal Data

## ABSTRACT

Depression recognition (DR) using facial images, audio signals, or language text recordings has achieved remarkable performance. Recently, multimodal DR has shown improved performance over single-modal methods by leveraging information from a combination of these modalities. However, collecting high-quality data containing all modalities poses a challenge. In particular, these methods often encounter performance degradation when certain modalities are either missing or degraded. To tackle this issue, we present a generalizable multimodal framework for DR by aggregating feature disentanglement and privileged knowledge distillation. In detail, our approach aims to disentangle homogeneous and heterogeneous features within multimodal signals while suppressing noise, thereby adaptively aggregating the most informative components for high-quality DR. Subsequently, we leverage knowledge distillation to transfer privileged knowledge from complete modalities to the observed input with limited information, thereby significantly improving the tolerance and compatibility. These strategies form our novel Feature Disentanglement and Privileged knowledge Distillation Network for DR, dubbed **Dis2DR**. Experimental evaluations on AVEC 2013, AVEC 2014, AVEC 2017, and AVEC 2019 datasets demonstrate the effectiveness of our Dis2DR method. Remarkably, Dis2DR achieves superior performance even when only a single modality is available, surpassing existing state-of-the-art multimodal DR approaches AVA-DepressNet by up to 9.8% on the AVEC 2013 dataset.

## CCS CONCEPTS

• **Applied computing → Health informatics**.

## KEYWORDS

Multimodal; Depression Recognition; Knowledge Distillation; Affective Computing

## 1 INTRODUCTION

Depression recognition (DR) has made significant progress with signals derived from facial images [46], speech audio [65], and language semantics [58]. Features extracted from these modalities have shown associations with depression disorder. Consequently,

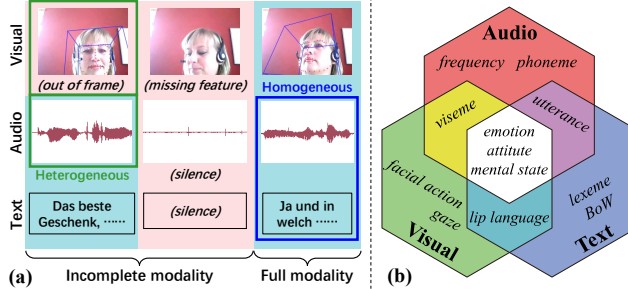

**Figure 1: (a) An illustration depicts the challenge of recognizing depression using incomplete multimodal data. Red blocks highlight missing information in the modalities. (b) A depiction showcases the heterogeneous content from different modalities, represented as primary RGB colors. Compound and white colors represent homogeneous information from combined modalities.**

recent studies have focused on leveraging the combination of multimodal signals to improve DR performance. In multimodal DR research, there is a particular emphasis on multimodal fusion, which has been proven effective in utilizing complementary information from multiple modalities rather than relying on a single modality [34, 36]. This evolution enables the possibility of non-contact depression screening, achievable through a multimedia file with facial video recordings and voice recordings during speech. However, real-world scenarios of DR often present the following challenges, which hinder further improvement of multimodal DR:

i) *Modality Degradation.* The information within a single modality is occasionally of varying quality. In some instances, certain content may be missing. For example, eye tracking data may be lost while other features remain intact. Alternatively, content may be compromised by external factors, resulting in degraded quality. For instance, head movements may disrupt the accuracy of facial feature extraction, while background noise can diminish speech feature quality. The interference is common in input signals and can inherently impact DR performance.

ii) *Modality Incompleteness.* Multimedia information frequently suffers from modality incompleteness, which may arise from feature detection failures, face out of frame, or speechless moment, resulting in the partial information existence. Such occurrences may manifest even during the data collection and training phases, and the absence of information can result in sparse content, and adversely impact the convergence of the model during training.

To address the aforementioned challenges, this study introduces a novel Feature Disentanglement and Privileged Knowledge Distillation Network for DR (**Dis2DR**). The Dis2DR framework aims to enhance the model's generalization across modalities by effectively balancing the utilization of both homogeneous and heterogeneous

multimodal information. This approach reduces the model's reliance on specific modalities and mitigates the impact of modality degradation during DR. During knowledge distillation, the teacher model leverages full input as privileged knowledge, while the student model characterizes complete distribution with partial input and learnable privileged knowledge, thus enabling it to learn effectively in such settings. This method is well-suited to address modality incompleteness challenges commonly encountered in DR settings. Specifically, this work comprises the following components:

i) Dis2DR encompasses a fundamental disentangling DR model and associated constraints for the disentanglement of multimodal information, with the objective of separating depression-related multimodal homogeneous and heterogeneous features, and modality noise from the input data. This disentanglement process augments the representation capacity of multimodal features while diminishing interference from irrelevant information in the DR task. Subsequently, depression-related features are projected onto a low-dimensional latent space, from which depression severity scores are predicted.

ii) The process of feature disentanglement heavily relies on interdependence across modalities, which can lead to a performance drop if certain modalities are degraded. Dis2DR addresses this by having the teacher model learn disentanglement using rich content under full modality input as privileged knowledge, then distilling this knowledge to the student model, which handles incomplete input. This ensures representations of disentangled features while learning modality-independent representations in student model, enhancing the generalization representation between modalities, resulting in robust DR performance in scenarios where input modalities are degraded or missing.

iii) The experiment demonstrates that feature disentanglement in Dis2DR enhances the homogeneity of inter-modality depression-related content while effectively separating heterogeneous information, both of which are important to DR. Moreover, utilizing privileged knowledge distillation enhances DR performance by improving the generalization of modalities, especially in scenarios involving incomplete modalities, even when only a single modality is available.

## 2 RELATED WORKS

### 2.1 Depression Recognition

In DR with single modality inputs, visual information typically includes facial Action Units, pose, and gaze [48]. Facial images [67] or video frames [35] are commonly used for end-to-end recognition. In audio analysis, acoustic feature-based approaches [32] are prevalent, often combined with deep learning models [37]. For text inputs, language features are preprocessed using word embeddings [58] or deep embeddings [50], with linguistic features directly reflecting emotion tendencies. Approaches for DR with multimodal signals typically entail combining multimodal inputs, utilizing a variety of handcrafted features [24], or features extracted by deep learning model [25]. Some approaches emphasize the importance of integrating text [45] and substantial fusion processing [62, 63] to achieve optimal performance. However, many multimodal methodologies rely heavily on manually crafted features, often overlooking

potential failures in feature detection, such as facial motion blurring or silent sections in speech. Such oversights could significantly affect the effectiveness of DR when certain contents or modalities are absent. Furthermore, subsequent research fails to explore the interplay between different modalities, limiting the comprehensive utilization of modality information by the model.

### 2.2 Feature disentanglement

Disentangled representation learning, as highlighted in previous research [6], aims to constrain features to disentangle independent factors within the data. A well-designed disentangled representation aligns with the semantic structure of the data. In facial recognition, typical disentanglement factors include identity, pose, and emotion information [30, 47]. In multimodal emotion recognition tasks, disentangling of multimodal features has been explored using techniques like graph distillation [29]. Typically, modality information is disentangled into modality-invariant and modality-specific subspaces [60]. However, the specific disentangling strategy employed varies significantly depending on the attributes of the task at hand. In the context of DR, our study is the first to propose a disentangled representation framework specifically tailored to depression disorder. This framework considers both homogeneous and heterogeneous depression-related multimodal features, marking a novel contribution to the field.

### 2.3 Incomplete Multimodal Learning

Incomplete multimodal learning, a critical area of research within multimodal machine learning, addresses scenarios where certain modalities degrade, a common issue in real-world settings. While one effective strategy involves identifying a low-dimensional subspace shared by all modalities, maximizing their correlation [2, 22, 56], this methodology may overlook the complementary nature of heterogeneous modalities, potentially leading to suboptimal outcomes. Instead, a more promising approach is to explicitly recover missing modalities using available ones. For example, deep models [10, 53] or cross-modality recovery strategies with cycle consistency loss [39, 64] can be employed for this purpose. However, many of these approaches require substantial full-modality data, which is often lacking in DR datasets due to their limited size and presence of missing modalities. Some methodologies involve utilizing main and complementary modalities for learning using privileged information [3, 7, 20], but subsequent approaches focus on treating entire modalities as privileged information, requiring high availability of the required modality during teacher model training. In the context of DR, modalities may be incomplete even within the training data, posing a challenge to the effective utilization of learning using privileged information. We aim to explore the application of privileged information in incomplete modalities for the DR task, marking a novel contribution to the field.

## 3 THE PROPOSED METHOD

The overall structure of Dis2DR is illustrated in Fig. 2 (a), which consists of a teacher model and a student model engaged in multimodal privileged knowledge distillation. In the teacher model, the input undergoes processing by the Incomplete Modality Interaction (IMI)

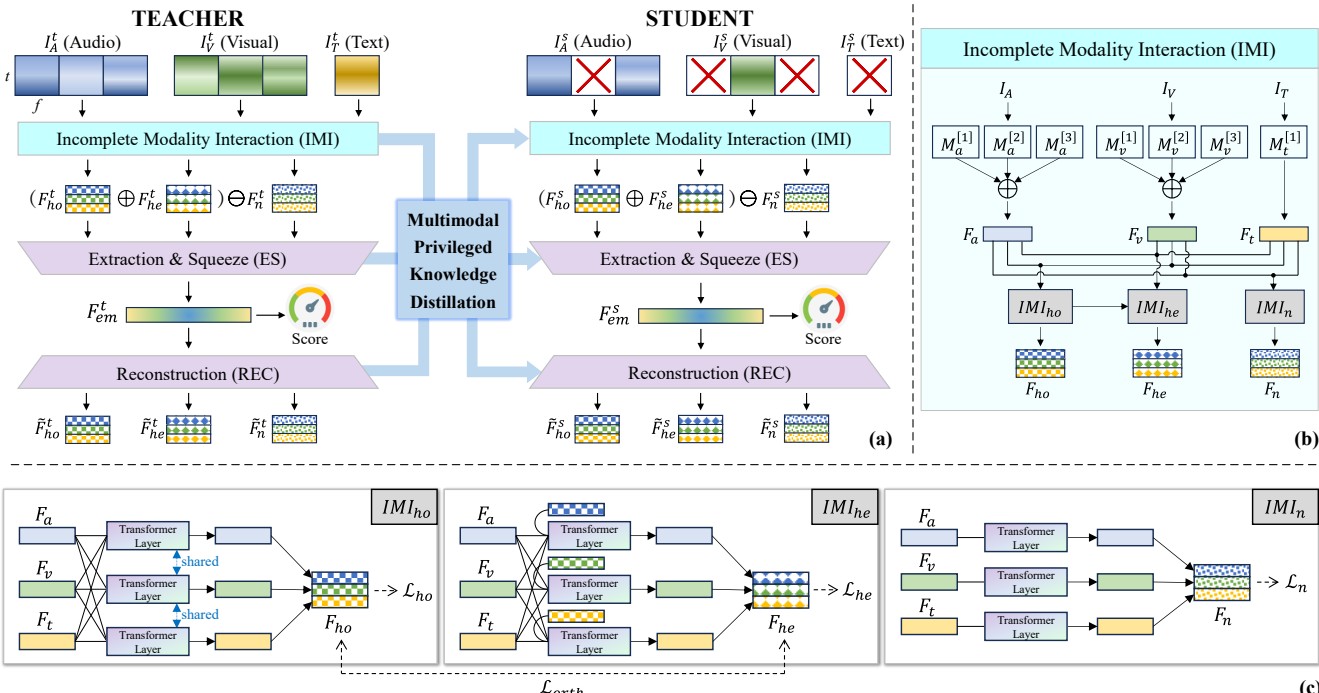

Figure 2: The overall architecture of Dis2DR. (a) The multimodal privileged knowledge distillation process between the teacher and student models. During inference, only partial inputs are used with the student model. (b) An illustration of the IMI model. (c) The pipeline of IMI sub-module for feature disentanglement.

module, responsible for disentangling. Through this process, homogeneous, heterogeneous, and noise multimodal representations are obtained by interacting across modalities and enforcing constraints imposed by the designed loss functions. Subsequently, the representation undergoes compression and reconstruction via an encoder-decoder structure comprising the extraction and squeeze (ES) layer and the reconstruction (REC) layer. The latent embeddings $F_{em}$ is then utilized for deriving the DR score. Leveraging the encoder-decoder structure facilitates the self-supervised pretraining of Dis2DR using a large multimodal dataset, thus mitigating the challenge of small DR datasets for deep learning.

The student model adopts an identical architecture to the teacher model, with the goal of minimizing disparities and aligning knowledge between the two models. This alignment facilitates seamless knowledge transfer from the teacher to the student. Subsequently, the teacher and student models engage in multimodal privileged knowledge distillation, ensuring the effective transmission of information. The student model then proceeds to learn disentangled representations under incomplete modalities, thereby improving the generalization of modality information.

In order to enable students to learn from missing or degraded multimodal data, we carefully prepare the data fed to the teacher network and the student network. The teacher network is trained on data with more comprehensive multimodal features, representing privileged knowledge. Meanwhile, the student network is exposed to data with less information, simulating real incomplete multimodal inputs, and benefiting from supervision provided by the teacher model. This setup enables the student to effectively adapt to scenarios where modalities are missing or degraded.

## 3.1 Structure of Teacher Model

The input of the DR model in Dis2DR consists of a combination of audio, visual, and text data. Specifically, $I_A^t, I_V^t, I_T^t$ represent the teacher model inputs from audio, visual, and text, respectively. Similarly, $I^s$ represent the student model inputs from each modality.

For training the basic DR model, it need to calculate the loss function of the following part: 1) the error of DR prediction $\mathcal{L}_s$ between the ground truth depression score and predicted one; 2) the loss function $\mathcal{L}_{vae}$ for the latent embeddings; 3) the loss of IMI $\mathcal{L}_{IMI}$ during feature disentangling. In the following, each loss functions are presented in detail.

For the $\mathcal{L}_s$, the sum of Mean Absolute Error (MAE) and Root Mean Square Error (RMSE) between the ground truth label $y$ and predicted score $\hat{y}$ from $N$ samples is calculated, as the goal of DR:

$$\mathcal{L}_s = \frac{1}{N} \sum_{i=1}^{N} |\hat{y}_i - y_i| + \sqrt{\frac{1}{N} \sum_{i=1}^{N} (\hat{y}_i - y_i)^2}, \quad (1)$$

The $\mathcal{L}_{vae}$ contains the reconstruction error between the input multimodal feature and the reconstructed feature. In formulation, $\mathcal{L}_{vae}$ can be written as:

$$\mathcal{L}_{vae} = \frac{1}{N} \sum_{i=1}^{N} (\tilde{F}_i - F_i)^2 - \frac{1}{2} \times (2log\sigma + 1 - \sigma^2 - \mu^2). \quad (2)$$

In the following section, we present the details of $\mathcal{L}_{IMI}$.

## 3.2 Incomplete Modality Interaction

In this case, $I_A \in \mathbb{R}^{t,f_a}$ consists of several audio feature sets with a total dimension of $f_a$ that are merged along the feature axis. The same applies to $I_V$, while $I_T$ comprises solely the text embedding as the only feature set. The IMI first extracts and standardizes the temporal series of the input with $M_a$, $M_v$ and $M_t$, representing the corresponding feature extraction layers. For $M_a$ and $M_v$, they have sublayers denoted as $M^{[i]}$ for processing the $i$-th feature set from the modality, and then aggregate the cumulative result.

The multimodal features can be processed using three IMI submodules to disentangle the homogeneous components $F_{ho}$, heterogeneous information $F_{he}$ and the noise counterpart $F_n$. The corresponding IMI submodules are denoted as $IMI_{ho}$, $IMI_{he}$, $IMI_n$, respectively. For simplicity of understanding, the illustration of IMI is shown in Fig. 2 (b), (c). In the IMI submodule, the Transformer block is used for feature extraction. In $IMI_{ho}$, all modalities utilize the same Transformer. Each modality contributes its primary representation and a modality token (using the first temporal dimension) to generate the multimodal feature $F_{ho}$. In $IMI_{he}$, each Transformer receives not only the primary multimodal representation but also the corresponding $F_{ho}$ specific to that modality. The $IMI_n$ module receives single-modality information and predicts by corresponding Transformers. The modality tokens of $F_{ho}$, $F_{he}$, and $F_n$ are selected and concatenated, as shown in Fig. 2 (c).

To constrain $F_{ho}, F_{he}, F_n$, specific loss functions are designed. Firstly, we propose the Depression Level Contrastive (DLC) loss, aiming to ensure consistent representation within the same depression levels and diverse representation across different levels. This is achieved by minimizing similarity for similar depression levels and maximizing it for distinct levels within each sample. The DLC loss between $N$ samples is formulated as:

$$\mathcal{L}_{dlc} = [(1 - |y_i - y_j|) \cos(F_i, F_j) \\ + |y_i - y_j| \max(\lambda_{dlc} \cdot |y_i - y_j| - \cos(F_i, F_j), 0)]_{i,j=1}^N, \quad (3)$$

where $y$ represents the corresponding normalized label of samples, $cos(\cdot)$ represents the cosine similarity. The margin $\lambda_{dlc}$ is set to 30.

For $F_{ho}$, the goal is to ensure that multimodal features contain homogeneous information. This entails each sample having a representation reflecting individual homogeneity in depression symptoms, accomplished by minimizing similarity between modalities in features from the same sample with an inter-sample constraint. The loss function $\mathcal{L}_{ho}$ combines this objective with the DLC loss, formulated as:

$$\mathcal{L}_{ho} = \mathcal{L}_{dlc} + \sum_{p,q \in \{a,v,t\}} \left[ \cos(F_{ho_i}^p, F_{ho_i}^q) \right]_{i=1}^N. \quad (4)$$

Similarly, $F_{he}$ should contain heterogeneous representations among different modalities within a depression sample, highlighting the diverse information across modalities for characterizing individuals with depression. The objective aims to maximize the similarity of multimodal features from one sample. The loss function $\mathcal{L}_{he}$ can be formulated as:

$$\mathcal{L}_{hes} = \mathcal{L}_{dlc} + \sum_{p,q \in \{a,v,t\}} \left[ \max(\lambda_{he} - \cos(F_{he_i}^p, F_{he_i}^q), 0) \right]_{i=1}^N, \quad (5)$$

where the margin $\lambda_{he}$ is set to 10.

To maximize the distinction between $F_{ho}$ and $F_{he}$, ensuring they represent features with minimal redundancy, the loss $\mathcal{L}_{orth}$ is designed to maintain their orthogonality, and can be expressed as:

$$\mathcal{L}_{orth} = \left( F_{ho_i} F_{he_i}^\top \right)_{i=1}^N. \quad (6)$$

For $F_n$, the $\mathcal{L}_n$ emphasizes the contrastive representation across modalities, ensuring that each modality has its own distinct noise representation, while remaining consistent within one modality regardless of the depression levels. The $\mathcal{L}_n$ is formulated as:

$$\mathcal{L}_n = \left[ \cos(F_{n_i}, F_{n_j}) \right]_{i,j=1}^N + \sum_{p,q \in \{a,v,t\}} \left[ \max(\lambda_n - \cos(F_{n_i}^p, F_{n_i}^q), 0) \right]_{i=1}^N, \quad (7)$$

where the margin $\lambda_n$ is set to 10.

Overall, the $\mathcal{L}_{IMI}$ can be presented as

$$\mathcal{L}_{IMI} = \mathcal{L}_{ho} + \mathcal{L}_{he} + \mathcal{L}_n + 0.1 \cdot \mathcal{L}_{orth}. \quad (8)$$

## 3.3 Multimodal Privileged Knowledge Distillation

We utilize the teacher model learned with full modalities information to provide privileged knowledge for the student model learned with incomplete information. Compared to directly fine-tuning the teacher with incomplete modality, distillation to the student can offer stable, highly accurate supervision during the student-training procedure under incomplete inputs, preventing the model from forgetting. We choose to distill the key parts of the model, which contain the IMI, ES, and REC layers.

We design the following loss function to distill privileged knowledge based on the teacher's DR performance. The lower the error of the teacher model on a particular sample, the higher the temperature to distill the corresponding knowledge to the student. For each layer, the distillation function is denoted as:

$$\mathcal{L}_{kd} = \sum_j \left[ \left( 1 - |y - \hat{y}^t| \right) \cdot |F_j^s - F_j^t| \right]. \quad (9)$$

$\hat{y}^t$ represents the output of the teacher, which is used as a temperature to control the strength of distillation, avoiding the influence of teacher misguidance. $F_j$ represents the feature obtained from the IMI, ES, and REC layers.

## 4 EXPERIMENTS

### 4.1 Datasets

In our experiments, we utilize the following datasets:
**AVEC 2013** [55]: Within this dataset, there are 150 video snippets featuring 82 distinct participants, specifically designated for the validation and testing stages of our research endeavor.
**AVEC 2014** [54]: Consisting of 300 video segments featuring contributions from 83 participants, both AVEC 2013 and AVEC 2014 capture interactions in a human-computer setting. The ratings, evaluated using the Beck Depression Inventory-II (BDI) [5], span from 0 to 63. Videos from these datasets are utilized for training, validation, and testing according to the official partitioning.
**AVEC 2017** [44]: In AVEC 2017, a collection of 189 samples from distinct individuals is involved. The severity of depression for each individual is gauged based on the self-reported PHQ-8 scores [28].

**Table 1: The depression score and corresponding level of the datasets.**

| BDI-II score | Level | PHQ-8 score | Level |
|---|---|---|---|
| 0−13 | Minimal | 0−4 | None |
| 14−19 | Mild | 5−9 | Mild |
| 20−28 | Moderate | 10−14 | Moderate |
| 29−63 | Severe | 15−19 | Moderately severe |
| | | 20−24 | Severe |

Video frames are dissected into various features, including Action Units (AU) and facial landmarks. Simultaneously, audio recordings are captured at a sample rate of 16kHz, with the AVEC 2017 dataset comprising extracted audio features such as formants and Fundamental Frequency (F0). Meanwhile the dialog transcript of each individual is recorded.

**AVEC 2019** [43]: AVEC 2019 constitutes an extension of AVEC 2017, encompassing a sample size of 275 instances. Notably, the facial landmark feature is omitted, while new deep features such as VGG and DenseNet features are incorporated into the AVEC 2019 dataset.

**CMDC** [68]: This dataset comprises 52 samples from healthy individuals and 26 samples from patients with depression. It encompasses transcripts, speech audio files, and, in some cases, records of facial visual features.

**VoxCeleb2** [9]: The VoxCeleb2 dataset is a large-scale speaker recognition dataset that contains over 1 million utterances attributed to 6,112 celebrities. These utterances are extracted from videos uploaded to YouTube. The dataset serves as a robust foundation for pretraining. In our study, we focus on extracting both facial and audio features from this dataset for pretraining the Dis2DR.

We utilize all the aforementioned datasets for model training, while the AVEC datasets are used for our evaluation and final testing. AVEC 2013 and AVEC 2014 are labeled with BDI-II, whereas AVEC 2017 and AVEC 2019 are labeled with PHQ-8. The corresponding criteria scores and depression levels are listed in Tab. 1. Initially, VoxCeleb2 is employed to train the encoder-decoder in the audio-visual task. Subsequently, we employ the joint set of CMDC and all the training sets of the AVEC datasets, normalizing the labels to a range of 0-1 based on their corresponding questionnaire responses. Due to the availability of official development and testing sets provided by the AVEC datasets, all ablation studies are conducted on their corresponding development set, while comparisons with state-of-the-art methods are performed on the testing set.

### 4.2 Preprocess

For all datasets, the audio, visual, and text data have been preprocessed and standardized for input into Dis2DR. Specifically, the utilized standard visual features include Histogram of Oriented Gradients (HOG) feature (if available), head pose, gaze, AUs, and facial landmarks with 68 points. The landmarks are aligned according to the center nose point and resized to a uniform size of 256 along the axis range. All the features are extracted from the facial images by OpenFace [4] or reorganized from the provided data in the dataset. For the audio modality, three sets of features are extracted, comprising the COVAREP [15] feature set, as well as Mel-Frequency

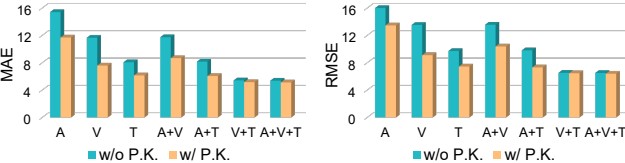

**Figure 3: The DR performance using audio (A), visual (V), text (T), and their combination as input modalities for Dis2DR w/o and w/ privileged knowledge (P.K.) is illustrated. The performance metrics are reported on the development set of AVEC 2014 datasets.**

Cepstral Coefficients (MFCC) and eGeMAPS feature set extracted by OpenSmile [17]. For the text modality, BERT [16] embeddings are extracted from the transcript as Dis2DR text modality input. For AVEC 2013 and AVEC 2014, the speaker's transcript is initially extracted from raw audio by multilingual Whisper [41], and then German BERT[1] is used to extract the embeddings. For AVEC 2017 and AVEC 2019, the English BERT model[2] is utilized. For CMDC, the Chinese BERT model[3] is employed.

### 4.3 Experiment Settings

In our experiments, the teacher model is firstly pre-trained on VoxCeleb2 to learn the audio-visual feature representation. In this step, the loss function is

$$\mathcal{L}^p = \mathcal{L}_{IMI} + \mathcal{L}_{vae}. \tag{10}$$

Then, the teacher is further trained with the mixed dataset comprising all AVEC datasets and CMDC dataset, with their sample labels standardized to a range [0, 1]. The training loss is

$$\mathcal{L}^t = 0.01 \cdot \mathcal{L}_{IMI}^t + \mathcal{L}_s^t + \mathcal{L}_{vae}^t. \tag{11}$$

As for the student model, it is initialized by copying the parameters from the teacher, which is frozen, and then is fine-tuned with the following loss function:

$$\mathcal{L}^s = 0.01 \cdot \mathcal{L}_{IMI}^s + \mathcal{L}_s^s + \mathcal{L}_{vae}^s + 0.001 \cdot \mathcal{L}_{kd}. \tag{12}$$

During privileged knowledge distillation in Dis2DR, the input of the student model is randomly masked along the temporal axis with stochastic position and length. Consequently, the teacher model continues to utilize full and complete data as input, which can be served as the privileged knowledge provider to the student.

All training is conducted on 2 RTX 3090 GPUs using the Adam optimizer with a learning rate of 0.0002 and a batch size of 16. The temporal length of the input is set to 600 frames, randomly sampled from the raw training signal. To address misaligned temporal sampling rates between audio and visual signals, we resample the clips. Text embedding utilizes the full embedding from a sample as the text input. During testing, the sample is divided into 10 uniformly sized pieces along the temporal length, each comprising 600 frames. The average score across these 10 clips is used as the model's prediction.

---

[1]German BERT: https://huggingface.co/dbmdz/bert-base-german-uncased
[2]English BERT: https://huggingface.co/google-bert/bert-base-uncased
[3]Chinese BERT: https://huggingface.co/google-bert/bert-base-chinese

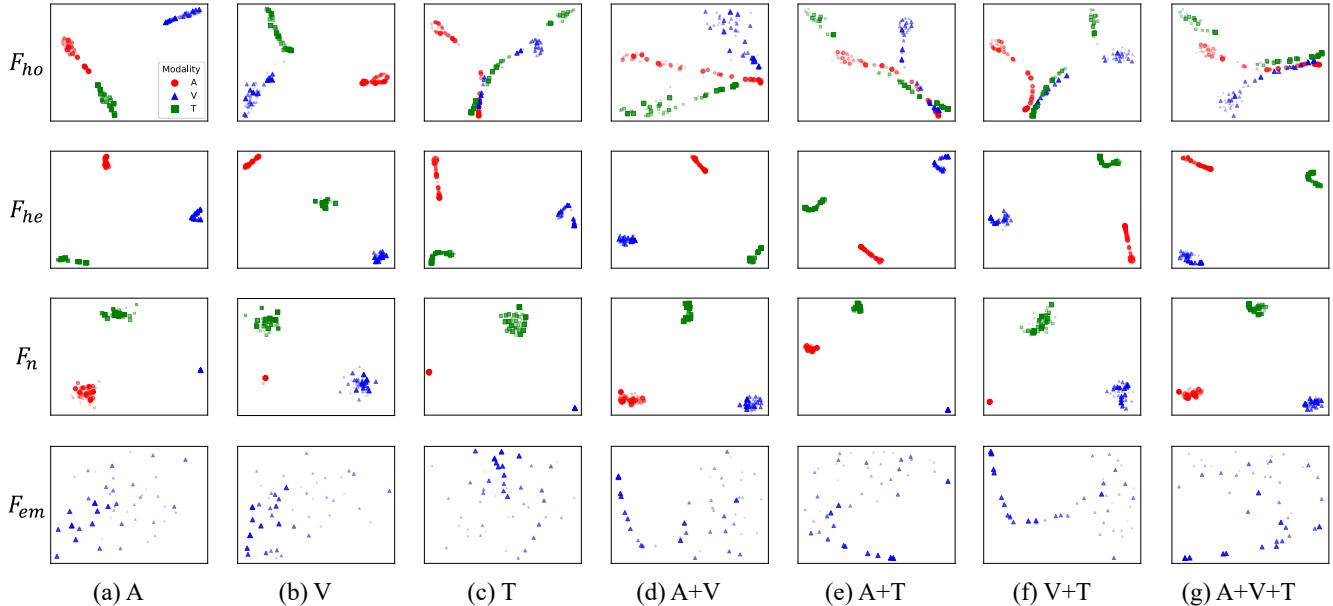

Figure 4: The t-SNE analysis displays the disentangled multimodal features from AVEC 2017. Each column presents the visualization results with the noted modalities available as inputs. A larger-sized dot with a thicker color represents a higher depression score for the corresponding sample.

## 4.4 Analysis of Privilleged Knowledge Distillation

We compare the performance of direct recognition using the student model before and after knowledge distilling, representing the existence and absence of privileged knowledge from Dis2DR for recognition from incomplete modality, highlighting the necessity of privileged knowledge distillation. The comparison results are presented in Fig. 3.

The results demonstrate a significant performance drop in incomplete modality DR when privileged knowledge is not utilized. Models with privileged knowledge consistently exhibit lower DR errors compared to those without privileged knowledge, as indicated by the yellow bars (with privileged knowledge) showing lower errors compared to the cyan bars (without privileged knowledge). Furthermore, comparing results with and without the text modality (T, A+T, V+T, A+V+T versus A, V, A+V), Fig. 3 illustrates that the performance drop is primarily associated with the absence of the text modality. Particularly, when the text modality is missing, the drop is more pronounced, followed by a noticeable decline in performance when the visual modality is absent. After privileged knowledge distillation, the imbalance in the dependency of DR on features has decreased. The absence of certain modalities does not lead to as significant a performance drop as observed without privileged knowledge. This emphasizes the crucial role of knowledge distillation in incomplete modality scenarios. It mitigates the heavy reliance on features across modalities, ensuring high availability even when certain modalities are degraded or missing.

## 4.5 Analysis of Disentangled Features

The student model trained in Dis2DR is tested with various types of inputs to examine the influence of different modalities on the features using t-SNE for analysis. As depicted in Fig. 4, the $F_{ho}$, $F_{he}$, $F_n$, $F_{em}$ with different inputs are displayed in rows. Each column corresponds to a specific input modality. The following trends can be inferred by the results: i) The visual and text modalities are deemed more crucial, substantially contributing to performance improvement. The clustering of $F_{ho}$ reveals that the absence of the text modality has a substantial negative impact on the clusters, followed by the influence of the visual modality. Moreover, according to the t-SNE results, particularly in cases where only the audio modality is available, the audio and text modalities exhibit high entanglement. This suggests that the features captured by $F_{ho}$ in the absence of text modality can be "implied" by the existing audio modality. ii) $F_{ho}$ can learn identical representations between modalities, exhibiting a clear trend that correlates with the degree of depression. In the multimodal case of $F_{ho}$ (the first row of Fig. 4), samples with higher degrees of depression form tight and closely clustered groups across modalities, whereas samples with mild depression exhibit more diverse representations, resulting in divergence between each modality. iii) $F_{he}$ consistently performs well in representing heterogeneous information between modalities. Even when a modality is absent, the model still "remembers" the feature and provides a proper representation of this modality. iv) $F_n$ consistently exhibits strong clustering and clear boundaries even when some modalities are missing. This indicates that the noise feature represents coherent information within the modality and remains independent across samples. v) The $F_{em}$ displays a

**Table 2: The DR performance when disabling the corresponding disentangled features in the student model of Dis2DR.**

|  | AVEC 2013 | | AVEC 2014 | | AVEC 2017 | | AVEC 2019 | |
|---|---|---|---|---|---|---|---|---|
|  | MAE↓ | RMSE↓ | MAE↓ | RMSE↓ | MAE↓ | RMSE↓ | MAE↓ | RMSE↓ |
| w/o $F_{ho}$ & $F_{he}$ | 7.90 | 9.82 | 7.98 | 9.97 | 5.17 | 6.11 | 5.33 | 6.20 |
| w/o $F_{he}$ | 7.10 | 9.21 | 6.97 | 9.10 | 4.98 | 5.78 | 5.01 | 6.25 |
| w/o $F_{ho}$ | 6.81 | 7.99 | 6.82 | 8.08 | 4.92 | 5.39 | 4.97 | 5.81 |
| w/o $F_n$ | 6.46 | 7.78 | 6.19 | 7.76 | 4.52 | 5.35 | 4.32 | 5.28 |
| **w/ $F_{ho}, F_{he}, F_n$** | **6.28** | **7.74** | **6.14** | **7.85** | **4.41** | **5.28** | **4.01** | **5.19** |

**Table 3: The utilized features for distillation and the corresponding DR performance of Dis2DR.**

|  | AVEC 2013 | | AVEC 2014 | | AVEC 2017 | | AVEC 2019 | |
|---|---|---|---|---|---|---|---|---|
|  | MAE↓ | RMSE↓ | MAE↓ | RMSE↓ | MAE↓ | RMSE↓ | MAE↓ | RMSE↓ |
| w/ $F_n$ | 7.94 | 9.85 | 7.52 | 9.73 | 5.72 | 6.89 | 4.45 | 5.87 |
| w/ $F_{em}$ | 7.27 | 9.26 | 7.35 | 9.61 | 5.72 | 6.91 | 4.32 | 5.61 |
| w/ $F_{ho}$ | 6.81 | 8.64 | 6.67 | 8.56 | 5.33 | 5.77 | 4.24 | 5.38 |
| w/ $F_{he}$ | 6.51 | 8.14 | 6.43 | 8.21 | 5.18 | 5.68 | 4.23 | 5.34 |
| **Dis2DR** | **5.75** | **7.04** | **5.12** | **6.40** | **3.66** | **4.44** | **3.39** | **4.18** |
| w/o $F_n$ | 6.38 | 7.91 | 6.27 | 7.97 | 5.12 | 5.46 | 4.19 | 5.33 |
| w/o $F_{em}$ | 6.37 | 7.86 | 6.24 | 8.04 | 5.12 | 5.48 | 4.08 | 5.35 |
| w/o $F_{ho}$ | 7.35 | 8.83 | 7.42 | 9.07 | 5.56 | 6.73 | 4.43 | 5.47 |
| w/o $F_{he}$ | 7.73 | 9.64 | 7.61 | 9.42 | 5.68 | 6.77 | 4.43 | 5.73 |

distinguishing trend between samples with low and high degrees of depression. Samples with higher depression degrees form clear and tight clusters, while those with lower degrees tend to exhibit more diffuse clusters. This trend is particularly appeared when the input modalities increased.

Furthermore, we conduct a quantitative study to assess the influence of disentangled features for DR performance. To examine the impact when the features from the IMI are not functioning, we disable the corresponding loss function calculation during training, allowing the features to lose their constraints and downgrade to normal features. The results listed in Tab. 2 indicate that $F_{he}$ contributes the most, followed by $F_{ho}$. Moreover, if both $F_{ho}$ and $F_{he}$ are unavailable, it results in the most significant performance degradation.

### 4.6 Analysis of Features for Distillation

To evaluate the effectiveness of privileged knowledge distillation on the disentangled features in Dis2DR, we conducted experiments to distill or disable the distillation of these features during training. The performance results are presented in Tab. 3. The upper rows represent Dis2DR with only the listed corresponding feature distilled, while the lower rows represent the listed corresponding feature being removed during distillation. It is evident that distillation of $F_{he}$ has the most significant impact on performance, followed by $F_{ho}$. Specifically, distillation of only $F_{he}$ leads to a substantial performance improvement, while the absence of distillation for $F_{he}$ results in a significant performance drop. A similar trend is observed for $F_{ho}$, with a smaller impact compared to $F_{he}$. In contrast, distillation on $F_n$ and $F_{em}$ has a lesser impact on performance. This conclusion aligns with the trends observed in Tab. 2. This trend also addresses the question of whether we can directly fine-tune the teacher model for DR. As the distilled composition of the network decreases, representing a reduction in privileged

**Table 4: Performance comparison when utilizing $\mathcal{L}_{orth}$.**

|  | AVEC 2013 | | AVEC 2014 | | AVEC 2017 | | AVEC 2019 | |
|---|---|---|---|---|---|---|---|---|
|  | MAE↓ | RMSE↓ | MAE↓ | RMSE↓ | MAE↓ | RMSE↓ | MAE↓ | RMSE↓ |
| w/o $\mathcal{L}_{orth}$ | 7.27 | 9.81 | 7.21 | 9.72 | 4.92 | 5.73 | 4.86 | 5.93 |
| w/ $\mathcal{L}_{orth}$ | 6.28 | 7.74 | 6.14 | 7.85 | 4.81 | 5.28 | 4.01 | 5.19 |

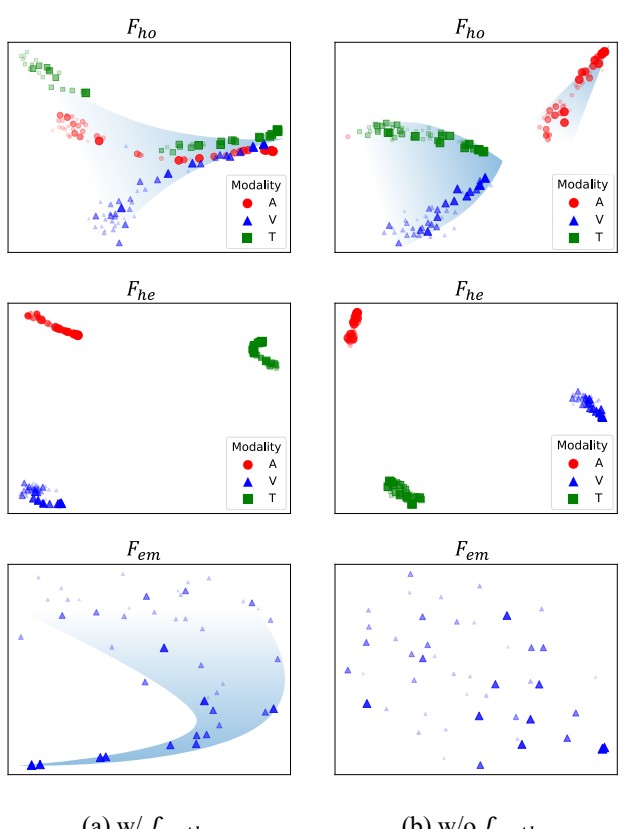

(a) w/ $\mathcal{L}_{orth}$ (b) w/o $\mathcal{L}_{orth}$

**Figure 5: The t-SNE analysis of $F_{ho}, F_{he}, F_{em}$ with the adoption of the orthogonality loss $\mathcal{L}_{orth}$ on the features from AVEC 2017. In the visualization, a larger-sized dot with a thicker color represents a higher depression score for the corresponding sample. Additionally, we illustrate the cluster tendency of the feature points using a blue gradient.**

knowledge, the training process tends to shift towards fine-tuning on incomplete modalities data. It's evident that the DR performance deteriorates compared to the Dis2DR method, further emphasizing the significance of privileged knowledge distillation.

### 4.7 Analysis of Orthogonality on Features

When designing the $F_{ho}$ and $F_{he}$, we utilize $\mathcal{L}_{orth}$ to maximize the distinction in information representation between the two features. We find that enforcing $\mathcal{L}_{orth}$ contributes to performance enhancement, as indicated in Tab. 4. Furthermore, we examine the involved features using t-SNE, as illustrated in Fig 5. It is evident that despite $F_{ho}$ and $F_{he}$ being constrained by the contrastive losses $\mathcal{L}_{ho}$ and

**Table 5: The comparison of Dis2DR and state-of-the-arts on testing set of AVEC datasets.**

**(a) AVEC 2013**

| Modality | Method | MAE↓ | RMSE↓ |
|---|---|---|---|
| A | AVEC 2013 Baseline [55] | 10.35 | 14.12 |
| | Two-Stage [31] | 10.88 | 14.49 |
| | PLSR [32] | 9.14 | 11.19 |
| | DCNN [21] | 8.20 | 10.00 |
| | Lp-norm [34] | 7.48 | 9.79 |
| | SAN-DCNN [66] | 7.38 | 9.65 |
| | MAFF [34] | **7.14** | 9.50 |
| | MFDS-VAN [37] | 7.29 | **9.43** |
| | **Dis2DR (A)** | 7.32 | 9.56 |
| V | AVEC 2013 Baseline [55] | 10.88 | 13.61 |
| | MAFF [34] | 7.32 | 8.97 |
| | LQGDNet [46] | 6.38 | 8.20 |
| | DepressNet [67] | 6.20 | 8.28 |
| | MDN [14] | 6.24 | 7.55 |
| | Behavior Primitives [48] | 6.16 | 8.10 |
| | STA-DRN [35] | 6.15 | 7.98 |
| | Depressioner [33] | 6.12 | **7.49** |
| | MSN [13] | **5.98** | 7.90 |
| | **Dis2DR (V)** | 6.04 | 7.50 |
| A-V | A-V System [26] | 9.09 | 11.19 |
| | MHH + PLSR [11] | - | 10.62 |
| | CCA [32] | 8.72 | 10.96 |
| | Kalman Filter [27] | 7.68 | 9.44 |
| | Two-Stage [31] | 6.75 | 8.29 |
| | MAFF [34] | 6.14 | 8.16 |
| | AVA-DepressNet [36] | 6.23 | 7.99 |
| | **Dis2DR (A-V)** | **6.12** | **7.97** |
| A-V-T | **Dis2DR** | **5.72** | **7.21** |

**(b) AVEC 2014**

| Modality | Method | MAE↓ | RMSE↓ |
|---|---|---|---|
| A | AVEC 2014 Baseline [54] | 10.04 | 12.57 |
| | PLSR [24] | 9.10 | 11.30 |
| | Fisher Vector [23] | 8.40 | 10.25 |
| | DCNN [21] | 8.19 | 10.00 |
| | Lp-norm [34] | 8.02 | 9.66 |
| | SAN-DCNN [66] | 7.94 | 9.57 |
| | MAFF [34] | 7.65 | **9.13** |
| | MFDS-VAN [37] | **7.33** | 9.44 |
| | **Dis2DR (A)** | 7.63 | 9.28 |
| V | AVEC 2014 Baseline [54] | 8.86 | 10.86 |
| | MAFF [34] | 6.43 | 8.60 |
| | DepressNet [67] | 6.21 | 8.39 |
| | LQGDNet [46] | 6.08 | 7.84 |
| | MDN [14] | 6.06 | 7.65 |
| | Depressioner [33] | 6.01 | 7.56 |
| | STA-DRN [35] | 6.00 | 7.75 |
| | Behavior Primitives [48] | 5.95 | 7.15 |
| | MSN [13] | **5.82** | 7.61 |
| | **Dis2DR (V)** | 5.92 | **7.09** |
| A-V | AVEC 2014 Baseline [54] | 7.89 | 9.89 |
| | Fusion System [38] | 8.99 | 10.82 |
| | CCA [27] | 7.69 | 9.61 |
| | GMM + ELM [59] | 6.31 | 8.12 |
| | PLSR + LR [25] | 6.14 | 7.43 |
| | M-BAM [8] | 5.78 | 7.47 |
| | MAFF [34] | **5.21** | 7.03 |
| | AVA-DepressNet [36] | 5.32 | 6.83 |
| | **Dis2DR (A-V)** | 5.45 | **6.61** |
| A-V-T | **Dis2DR** | **5.20** | 6.65 |

**(c) AVEC 2017**

| Modality | Method | MAE↓ | RMSE↓ |
|---|---|---|---|
| A | AVEC 2017 Baseline [44] | 5.72 | 7.78 |
| | CNN-GAN [57] | 7.32 | 8.56 |
| | AFN [40] | 5.67 | 6.55 |
| | LLD + Fisher Vector [51] | 5.30 | 6.34 |
| | LSTM [1] | 5.13 | 6.50 |
| | Random Forest [52] | 5.22 | 6.17 |
| | HATN [65] | 4.28 | 5.66 |
| | MFDS-VAN [37] | **4.27** | **5.34** |
| | **Dis2DR (A)** | 4.88 | 5.60 |
| V | AVEC 2017 Baseline [44] | 6.12 | 6.97 |
| | HOG-PCA [49] | 4.89 | 6.23 |
| | FDR + LDA [42] | 4.64 | 5.98 |
| | **Dis2DR (V)** | **4.51** | **5.88** |
| A-V | AVEC 2017 Baseline [44] | 5.66 | 7.05 |
| | AVA-DepressNet [36] | **4.62** | 5.78 |
| | **Dis2DR (A-V)** | 4.69 | **5.49** |
| A-V-T | ANEW + GSR [12] | 5.30 | 6.52 |
| | DCNN-DNN [61] | 5.16 | 5.97 |
| | A-V-T Hybrid [62] | 4.36 | 5.40 |
| | **Dis2DR** | **4.28** | **5.33** |

**(d) AVEC 2019**

| Modality | Method | CCC↑ | MAE↓ | RMSE↓ |
|---|---|---|---|---|
| A-V | AVEC 2019 Baseline [43] | 0.111 | - | 6.37 |
| | **Dis2DR (A-V)** | **0.514** | **4.32** | **5.35** |
| A-T | BERT-CNN + GCNN [45] | 0.403 | - | 6.11 |
| | MS-TDCNN [18] | 0.430 | 4.39 | 5.91 |
| | **Dis2DR (A-T)** | **0.467** | **4.28** | **5.34** |
| A-V-T | CubeMLP [50] | 0.583 | 4.37 | - |
| | Hierarchical Bi-LSTM [63] | 0.442 | - | 5.50 |
| | MFM-Att [19] | - | - | **5.17** |
| | **Dis2DR** | **0.608** | **4.21** | 5.28 |

$\mathcal{L}_{he}$, the absence of the orthogonal constraint results in suboptimal differentiation between features (as shown in the clusters in the first row). Without $\mathcal{L}_{orth}$, $F_{ho}$ exhibits numerous heterogeneous representations, which should have depicted homogeneous information rather than heterogeneous, as evidenced by the clusters separating into distinct groups. Additionally, the $F_{em}$ is not well-clustered and demonstrates a weak correlation with depression levels, attributable to the suboptimal representations of $F_{ho}$ and $F_{he}$ when $\mathcal{L}_{orth}$ is not enforced.

## 4.8 Comparison with the State-of-the-Arts

We compare our Dis2DR with state-of-the-art approaches in both single-modal and multimodal cases. The performance is presented in Tab. 5. An additional metric, Concordance Correlation Coefficient (CCC), is compared in AVEC 2019, which denotes the correlation between the predictions and ground truth. It is formulated as follows:

$$\text{CCC} = \frac{2\rho_{x,y}\sigma_x\sigma_y}{\sigma_x^2 + \sigma_y^2 + \left(\mu_x - \mu_y\right)^2}, \tag{13}$$

where $\sigma_x$ and $\sigma_y$ are the standard deviations, $\mu_x$ and $\mu_y$ are the mean values. $\rho_{x,y} = \text{cov}(x,y)/\sigma_x\sigma_y$ where cov is the covariance.

Overall, our Dis2DR achieves the best audio-visual-text multimodal DR performance across almost all criteria and across all the AVEC datasets. Even when only a subset of modalities is used, our Dis2DR demonstrates highly competitive performance compared to audio-visual and audio-text approaches on AVEC 2019. In terms of

single modality comparisons, our Dis2DR even surpasses most single audio or visual modality approaches. On AVEC 2013 and AVEC 2014 datasets, the audio-visual approach MAFF [34] achieves competitive performance compared to our Dis2DR framework. However, MAFF experiences significant performance degradation in the visual modality when considering single-modality approaches. In contrast, our Dis2DR framework maintains robust performance, approaching state-of-the-art performance levels even in the visual modality.

## 5 CONCLUSION

In this study, we introduce Dis2DR, an innovative framework that combines disentangled depression-related multimodal features with a privileged knowledge distillation paradigm for incomplete multimodal DR. Through the disentanglement of features into homogeneous, heterogeneous, and noise representations, Dis2DR effectively extracts depression-related features from both modality-specific and modality-invariant content, capturing crucial information and suppressing irrelevant content across various modalities. Furthermore, our privileged knowledge distillation approach leverages missing content as privileged knowledge, facilitating the generalization of modality information and improving performance in incomplete multimodal scenarios. Experimental results demonstrate Dis2DR's state-of-the-art performance in DR across full audio-visual-text modalities, while remaining competitive even with single or double modalities inputs on established benchmarks.

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
