# OpenReview forum: "Disentangled-Multimodal Privileged Knowledge Distillation for Depression Recognition with Incomplete Multimodal Data"
_acmmm.org/ACMMM/2024/Conference — MM2024 Poster_

### Official Review · Reviewer_oHNS · 2024-05-21

**Rating:** 3
**Confidence:** 3

**Summary:**

This paper introduces a novel incomplete multimodal learning method for depression recognition called Dis2DR. To solve the uncertain missing modality issues, feature disentanglement and privileged knowledge distilling are applied. Some experiments are conducted to verify the effectiveness.

**Strengths:**

1. This writing is generally clear and easy to understand.
2. The figures and tables are properly organized.
3. The experimental analysis is very comprehensive and enlighting.
4. Idea of the first contribution makes sense, and its ablation experiments meet the expectation.

**Limitations:**

1. The second contribution seems overlapped with previous works. In research line of incomplete multimodal learning, the learning paradigm of knowledge distillation is common and have already employed by previous works[1, 2, 3]. These existing methods use teacher models (based on complete data) to transfer knowledge to student models (based on incomplete data), which is particularly similar to the second contribution of this manuscript. The author needs to clarify the differences between the proposed model and existing distilling based incomplete multimodal learning models, include these methods in the related work for discussion, and compare them in the experimental section (if the data allows).

2. The motivation of first contribution is less convincing. Considering the incomplete multimodal data scenario, it is impractical to disentangle homogeneous and heterogeneous features from an non-existing modality. However, the authors fail to clarify the reasons for employing feature disentanglement under incomplete multimodal scenarios.

3. The performance contribution seems minor. In table 5, it is worth noting that the performance gain is minor. In some cases, Dis2DR model performs worse than other models. The authors need to analyze these phenomena clearly.


References:

[1] Fan Xu, Pinyun Fu, Qi Huang, Bowei Zou, AiTi Aw, and Mingwen Wang. 2023. Leveraging Contrastive Learning and Knowledge Distillation for Incomplete Modality Rumor Detection. In Findings of the Association for Computational Linguistics: EMNLP 2023, pages 13492–13503, Singapore. Association for Computational Linguistics.

[2] Wei, S., Luo, C., & Luo, Y. (2023). MMANet: Margin-aware distillation and modality-aware regularization for incomplete multimodal learning. In Proceedings of the IEEE/CVF Conference on Computer Vision and Pattern Recognition (pp. 20039-20049).

[3] Qi Wang, Liang Zhan, Paul Thompson, and Jiayu Zhou. 2020. Multimodal Learning with Incomplete Modalities by Knowledge Distillation. In Proceedings of the 26th ACM SIGKDD International Conference on Knowledge Discovery & Data Mining (KDD '20). Association for Computing Machinery, New York, NY, USA, 1828–1838. https://doi.org/10.1145/3394486.3403234

**Suitability:**

3

---

### Official Review · Reviewer_Wt96 · 2024-05-26

**Rating:** 4
**Confidence:** 3

**Summary:**

This paper introduces a framework named Dis2DR, designed for depression recognition using incomplete multimodal data, leveraging disentangled feature representation and privileged knowledge distillation. Dis2DR effectively manages the challenges posed by incomplete or degraded modalities by separating depression-related features into homogeneous, heterogeneous, and noise components, and enhancing model generalization through knowledge transfer from a fully-informed teacher model to a partially-informed student model. The framework demonstrated satisfactory performance on multiple depression datasets.

**Strengths:**

(1) The writing of this paper is clear.

(2) Dis2DR achieves the best or comparable on multiple depression datasets, showing effectiveness and robustness.

(3) The visualizations and experimental analyses are comprehensive.

**Limitations:**

(1) What are the advantages of the feature factorization module proposed in this paper compared to existing feature decoupling methods used in multimodal sentiment analysis?
[1] Factorized Contrastive Learning: Going Beyond Multi-view Redundancy. NeurIPS2023.
[2] Quantifying & Modeling Multimodal Interactions: An Information Decomposition Framework. NeurIPS2023.
[3] Learning Modality-Specific and -Agnostic Representations for Asynchronous Multimodal Language Sequences. MM2022.
[4] Disentangled Representation Learning for Multimodal Emotion Recognition. MM2022.

(2) Decoupled noise features are used for depression recognition, which may limit the effectiveness of the method.

(3) This paper adopts multiple depression datasets for pretraining, which may be unfair in comparison to other methods.

(4) This paper requires pretraining on VoxCeleb2, followed by pretraining on multiple depression datasets, and finally fine-tuning on the target depression dataset. The entire process is cumbersome and challenging.

(5) There are some minor writing issues or typos. For example, undefined symbols are present in Equation 2. loss_{he} in Equation 5.

**Suitability:**

3

---

### Official Review · Reviewer_wkbZ · 2024-05-29

**Rating:** 1
**Confidence:** 4

**Summary:**

This paper proposes Dis2DR, which utilizes feature disentanglement and privileged knowledge distillation to achieve depression recognition under incomplete multimodal data.

**Strengths:**

* The authors realize the extraction of distinct types of features through feature entanglement to adaptively aggregate information.

*  The authors utilize knowledge distillation to achieve knowledge transfer from complete modality to missing modality, thus improving the robustness of the framework.

*  The performance is good on some datasets.

**Limitations:**

*  Insufficient innovation: the proposed feature untangling mechanism is not significantly improved compared to previous works [1][2][3], and thus is not novel enough.

[1] Hazarika, D., Zimmermann, R., & Poria, S. (2020, October). Misa: Modality-invariant and-specific representations for multimodal sentiment analysis. In Proceedings of the 28th ACM international conference on multimedia (pp. 1122-1131).

[2] Yang, D., Huang, S., Kuang, H., Du, Y., & Zhang, L. (2022, October). Disentangled representation learning for multimodal emotion recognition. In Proceedings of the 30th ACM International Conference on Multimedia (pp. 1642-1651).

[3] Guo, W., Huang, H., Kong, X., & He, R. (2019, October). Learning disentangled representation for cross-modal retrieval with deep mutual information estimation. In Proceedings of the 27th ACM International Conference on Multimedia (pp. 1712-1720).

* Incomplete setup of missing modality: this paper only considers the case that the entire modality is missing but ignores the case that some of the features within the modality are missing.

* How to represent the features of the missing part of the modality? How to generate incomplete multimodal data for use as input into the student network? The paper lacks detailed descriptions of the aforementioned.

* The experimental results are not good enough and the proposed framework does not significantly outperform baselines.

* How to determine the values of margin $\lamda_{dlc}$, margin $\lamda_he$ and margin $\lamda_{n}$, as well as the value of the weighting coefficient of \mathcal{L}_{orth}? Lack of relevant ablation experiments.

* In terms of the presentation quality and clarity of the paper, many math notations presented have no detail explanation and good to add to notation table. This will help readers better appreciate research findings of this paper. For example, the representation of feature $F_i$ is not clear.

**Suitability:**

3

---

### Meta-Review · Program_Chairs · 2024-07-14

**Recommendation:** Accept (Poster)
**Confidence:** 3

**Metareview:**

this paper investigates the problem of multi-modal depression recognition. specifically, to deal with the missing information and the homogeneous modalities, the paper introduces a method called Dis2DR. via disentangling homogeneous and heterogeneous features and privileged knowledge distillation, the proposed method achieves competitive results on multiple benchmarks.

this paper received initial ratings of R, BA, BR. on the positive side, the paper was well recognized for separating the homogeneous features, good experimental performance, visualization and discussion, and its writing. on the other hand, reviewers were concerned with the experimental settings, missing ablations, need for more discussion and comparison with related works, and the motivation. during rebuttal, the authors provided feedback on the mentioned issues, and addressed many of the concerns. two reviewers upgraded their final ratings to BA and BA, whereas one reviewer did not provide their final ratings.

since the AC was not able to finish the meta-review in time, the PC stepped up and carefully went through all the reviews and author feedbacks. after consideration, the PC recommends Accept. the authors are strongly encouraged to update the manuscript based on the rebuttal and further address issues raised by reviewers.